# Cancer Stem Cells and Nucleolin as Drivers of Carcinogenesis

**DOI:** 10.3390/ph14010060

**Published:** 2021-01-13

**Authors:** Laura Sofia Carvalho, Nélio Gonçalves, Nuno André Fonseca, João Nuno Moreira

**Affiliations:** 1CNC—Center for Neurosciences and Cell Biology, Center for Innovative Biomedicine and Biotechnology (CIBB), Faculty of Medicine (Polo 1), University of Coimbra, Rua Larga, 3004-504 Coimbra, Portugal; laurasofiacarvalho@gmail.com (L.S.C.); nelio.goncalves@cnc.uc.pt (N.G.); nuno.fonseca@cnc.uc.pt (N.A.F.); 2TREAT U, SA—Parque Industrial de Taveiro, Lote 44, 3045-508 Coimbra, Portugal; 3UC—University of Coimbra, CIBB, Faculty of Pharmacy (FFUC), Pólo das Ciências da Saúde, Azinhaga de Santa Comba, 3000-548 Coimbra, Portugal

**Keywords:** tumor heterogeneity, drug resistance, cancer stem cells, nucleolin, targeted therapies, epithelial-to-mesenchymal transition

## Abstract

Cancer, one of the most mortal diseases worldwide, is characterized by the gain of specific features and cellular heterogeneity. Clonal evolution is an established theory to explain heterogeneity, but the discovery of cancer stem cells expanded the concept to include the hierarchical growth and plasticity of cancer cells. The activation of epithelial-to-mesenchymal transition and its molecular players are widely correlated with the presence of cancer stem cells in tumors. Moreover, the acquisition of certain oncological features may be partially attributed to alterations in the levels, location or function of nucleolin, a multifunctional protein involved in several cellular processes. This review aims at integrating the established hallmarks of cancer with the plasticity of cancer cells as an emerging hallmark; responsible for tumor heterogeneity; therapy resistance and relapse. The discussion will contextualize the involvement of nucleolin in the establishment of cancer hallmarks and its application as a marker protein for targeted anticancer therapies

## 1. Introduction

Cancer incidence is increasing and has become one of the leading causes of death and morbidity worldwide [1]. In 2018, 18.1 million new cancer cases and 9.6 million cancer-related deaths were registered, and a 20% risk of cancer development before the age of 75 [2].

Over the years, the concept of cancer has evolved towards the current understanding as a complex and heterogeneous disease, whose cells acquire a set of key properties (the hallmarks of cancer) through paracrine interaction, the tumor microenvironment and even with the immune system [3]. The heterogeneity is associated with tumor progression, therapy resistance and subsequent relapse [4]. Clonal evolution of cancer cells has long been presented and accepted as a cause of heterogeneity [5]. However, the identification of tumor cells with stem-like features—the cancer stem cells (CSCs)—has introduced a new level of complexity, shifting the understanding on tumor growth and development from the purely clonal expansion towards a hierarchical organization of cancer cells [6]. Cell plasticity represents another piece of complexity, which relates with the ability to transit between a stem-like phenotype and a more differentiated state, or vice-versa [7]. Such property is also present in CSC, in a process mediated by the epithelial-to-mesenchymal transition [8]. Furthermore, CSCs have been shown to reside in the vicinity of the tumor vasculature, producing proangiogenic factors, and further expand under hypoxia [9]. All this could represent a complex adaptive mechanism, further supporting metastization and therapy resistance, ultimately reinforcing the importance of CSC in the tumor phenotype, and consequently as relevant therapeutic targets.

In this respect, nucleolin, a nucleolar protein, has several identified roles in essential intracellular pathways including transcription and translation, cell cycle and division, cell survival and differentiation [10]. Due to this multifunctional behavior, dysregulation of nucleolin was implicated in tumorigenesis and tumor maintenance. In this respect, relocation of nucleolin to the cell membrane and its overexpression were identified in cancers from diverse histological origins [11,12,13]. Consequently, nucleolin has been studied as a target for anticancer therapies [14,15].

In this work, an overview of already established hallmarks of cancer will be performed in the context of emerging ones: stemness and plasticity of cancer cells. In this respect, the involvement of nucleolin in processes whose dysregulation leads to cancer hallmarks, including CSC-associated processes, will be discussed and further integrated in its application as a target for anticancer therapies.

## 2. The Established Hallmarks of Cancer

Tumorigenesis is a multistep process that transforms normal cells (phenotypically and functionally diverse), via a series of premalignant states, into malignant and highly of invasive cancers [16]. Healthy cells go through mutations over time potentially activating genes with oncogenic capacity (oncogenes) and/or loss of function of other several key genes, such as tumor suppressor genes. Epigenetic studies have also recently highlighted novel cues underlying the development and maintenance of cancer [17]. These genetic and epigenetic alterations in multiple sites of the genome may drive a step by step gain of growth advantages, leading to a progressive transition towards malignancy [16].

Hanahan and Weinberg (2000) [16] proposed that most cancers, if not all, should acquire a set of six functional alterations during their development that collectively determine malignant growth. These functional alterations were named the hallmarks of cancer, and included: (1) self-sufficiency in growth signals; (2) insensitivity to growth-inhibitory signals; (3) evasion of programmed cell death (apoptosis); (4) limitless replicative potential; (5) sustained angiogenesis and (6) tissue invasion and metastases, which might vary chronologically and mechanistically, depending on the cancer histological origin.

In 2011, the same authors [3] revisited the proposed hallmarks and included new ones based on the current view of cancer as a complex dynamic tissue that depended on the interaction between cancer cells and the surrounding microenvironment. Two enabling characteristics were proposed, including the genomic instability that generates random mutations, likely driving to hallmark capabilities, and the inflammatory state of premalignant or malignant lesions, driven by immune cells that can promote tumor progression. Additionally, two additional hallmarks were proposed, namely, the reprograming of the cellular metabolism to support continuous cell growth and proliferation, and the ability of cancer cells to evade the immune system surveillance.

There are different examples supporting modulation of the immune system and of how the hallmarks may be connected. For example, melanoma-derived exosomes seemed to have a role in the establishment of the metastatic niche, upon education and mobilization of bone marrow-derived cells that promoted a provasculogenic phenotype and induced vascular leakiness, favoring evasion and infiltration [18]. Liver Kupfer cells uptake pancreatic ductal adenocarcinoma exosomes containing macrophage migration inhibitory factor (MIF), thus leading to secretion of transforming growth factor beta (TGF-β). TGF-β stimulates pancreatic stellate cells to produce fibronectin, promoting the arrest of bone marrow-derived macrophages and ultimately leading to premetastatic niche formation [19].

Alternatively, cancer cell heterogeneity also favors tumor development, further supporting the hallmarks of cancer. However, one could rather think that cancer cell heterogeneity arises solely from stochastic mutations, without affecting the tumorigenic competences of the cells. However, the demonstration that not all cancer cells are able to generate tumors in immunocompromised mice [20], suggested otherwise. In fact, growing evidence has suggested that a subpopulation of cancer cells with a stem-like phenotype might strongly contribute to this heterogeneity and the underlying hierarchical nature tumors, altogether favoring hallmark traits [20,21].

## 3. Cancer Stem Cells—Another Layer of Complexity

Stem cells (SCs) are undifferentiated cells present in all stages of life (from the embryonic stage until adulthood) with the capacity to differentiate into several cells with the capacity to build mature adult organs and/or of tissue regeneration. SCs are characterized by highly proliferative rates (self-renewal) and clonality [22].

The identification of cells with enhanced tumorigenic potential, within certain types of cancers, simultaneously presenting similar characteristics as those presented by SC, gave rise to the concept of CSC [6,20,21]. CSC are defined as a population of cells within the tumor microenvironment that are able to self-renew for self-maintenance of the population, and to differentiate into every tumor cell type, thus sustaining malignant growth [23]. Accordingly, this concept has been accommodated in the models of tumorigenesis in an attempt to explain cancer cell heterogeneity in the context of tumor growth and disease relapse.

### 3.1. Models of Tumorigenesis—A New Paradigm Driven by CSC

Classically, tumor development has been explained by the stochastic model, which proposes that all cancer cells are biologically equivalent and equally able to initiate tumorigenesis, varying in behavior uniquely due to stochastic extrinsic and intrinsic influences (Figure 1a) [23]. This model relies on the concept that an adult somatic cell suffers sequential genetic mutations and subsequently undergoes clonal expansion, originating a hyperproliferative tumor from multiple clonal evolutions [24]. However, the relatively recent concept of CSC allowed the establishment of a hierarchical model of tumorigenesis to explain tumor emergence, maintenance and heterogeneity. According to this model, CSCs are responsible for initiating, maintaining and seeding the tumor (Figure 1b).

Studies from Dick and colleagues [6,21] were the first to support the hierarchical model of tumorigenesis by showing that sorted CD34^+^/CD38^-^, but not CD34^+^/CD38^+^ and CD34^−^, human acute myeloid leukemia (AML) cells injected in mice could give rise to leukemia. Later, the work of Al-Hajj and colleagues [20] brought the concept of CSC to solid tumors by specifically identifying CD44^high^/CD24^low^ as tumorigenic breast cancer cells, able to proliferate and initiate tumors in vivo. Thenceforth, several reports have identified distinct cancer cell populations with the ability to self-renew and to initiate and maintain the tumor in several other organs, such as ovary [25], lung [26], skin [27], thyroid [28] or sarcoma [29], in agreement with the CSC concept and the hierarchical growth of tumors. The identification by in vivo lineage tracing of Lgr5+ cryp stem cells as the cells of origin of intestinal cancer was a landmark contribution to the hierarchical stemness concept in cancer [30,31]. Yet, despite the presence of bipotent stem cells, including Lgr5+ subsets, contributing to maintenance of the mammary gland [32], their clear involvement on the origin of breast cancer remains to be confirmed.

Nevertheless, the existence of cells with activated stemness programs have a direct negative impact on the tumor immune system, the latter often supporting the CSC niche. Indeed, tumor-associated macrophages (TAMs), skewed to an M2-like phenotype, support tumor growth by promoting aberrant angiogenesis and by suppressing the immune system [33,34]. However, their relevance to CSC niche was poorly known until recently. It has been shown that breast CSC niche is supported by TAM signaling mediated by their interaction through the CSC-overexpressed CD90 (Thy1) surface signaling molecule [35]. Such an interaction was shown to facilitate the tumorigenic process of breast CSC, while essential to maintain their mesenchymal/stem-like state. Further confirmation of the stemness/immune response relation in solid tumors was provided by integrated gene-expression analysis. While predicting patient survival, stemness activation signatures negatively correlated with immune cell infiltration, which was further confirmed by its association with immunologically cold cancers, intratumoral heterogeneity and overexpression of immunosuppressive pathways [36].

It is thus apparent that stemness, under the paradigm of CSC, may be the underlying foundation of many cancers and the hindering of the intrinsic immune response, rendering their clear identification an utmost medical need. Unfortunately, despite the Lgr5 examples above, a single universal CSC marker is yet to be determined owing to their genetic and epigenetic controlled evolution [37] or the occurrence of Darwinian selection of clones enabling selective or polyclonal engraftments [38]. However, putative CSC may be identified by surface markers (Table 1) or functional assays. Accordingly, CSC properties may be evaluated using assays like the ability to form spheres in suspension (property of stem cells in culture), expel dyes like Hoechst (through the overexpression of ABC transporters, originating the so-called CSC-enriched “Side Population”) or limiting dilution xenotransplantation in mice [39]. Further evaluation of the mRNA levels of pluripotency markers such as NANOG, SOX2 or OCT4 is often used [40]. Aldehyde dehydrogenase (ALDH) is highly expressed in CSC and is very often used as a marker [41].

Recently, the recognition of plasticity as a characteristic of cancer cells, enabling the shift between a well-differentiated state and undifferentiated stem-like phenotype [50], made possible the combination of the stochastic and the hierarchical models, which are not mutually exclusive [7], yet simultaneously challenging the identification of CSC populations. Accordingly, in addition to the differentiation capacity of a CSC into any non-stem cancer cell (non-SCC), a non-SCC can, in turn, transit into an undistinguishable and tumorigenic CSC phenotype [7] (Figure 2). The integration of cells in specific niches secreting factors promoting stemness seems to be important for the maintenance of a CSC-like phenotype. Separation of daughter cells from that environment may commit them to differentiation [51].

The recognition of epithelial-to-mesenchymal transition (EMT) and its inverse process mesenchymal-to-epithelial transition (MET) are mechanisms largely attributed to plasticity in cancer cells, and potentially underlying the observations above [24], as their activation modulates tumorigenesis [52].

### 3.2. Epithelial-to-Mesenchymal Transition

Despite a natural homeostatic mechanism of great relevance for embryogenesis, tissue regeneration and organ fibrosis, EMT has also been implicated in the establishment of tumorigenesis and metastasis. During EMT, an epithelial cell undergoes a morphological transformation acquiring a more mesenchymal phenotype. This process is characterized by a loss of epithelial markers (e.g., E-cadherin, occludin and cytokeratin), gain of mesenchymal markers (e.g., fibronectin, β-catenin and N-cadherin), loss of apical-basal polarization and stable cell–cell adhesions, resulting in enhanced migratory capacity [53].

During tumorigenesis, cells undergoing EMT are key mediators of improved migration, invasiveness and establishment of metastasis. The process may be induced by activation or dysregulation of oncogenic pathways (e.g., TGF-β, EGF, NF-κB and Wnt), hypoxia-induced expression of HIF-1α, or factors produced by the tumor microenvironment [54]. For instance, factors secreted by tumor-associated macrophages (TAMs) such as TGF-β [55] or IL-10 [56] can induce EMT in solid tumors. Additionally, cancer-associated fibroblasts (CAFs) promote EMT either by inducing EMT-promoting transcription factors, activating intracellular pathways or upon modulating methylation [57].

One of the most widely studied markers in the context of EMT is the reduction of E-cadherin expression. Meta-analysis studies showed that lower levels of E-cadherin expression are correlated with cancer patients’ poorer prognosis and overall survival [58,59]. It is now known that EMT-promoting transcription factors Slug and Snail repress E-cadherin expression by binding to specific E-box elements in its promotor [60,61]. More recently, demonstration of miR-221 upregulation by Slug [62] and the repression of E-cadherin promotor by Zeb1, another EMT transcription factor [63], have been pointed as additional mechanisms to downregulate E-cadherin. Hence, EMT programs-mediated modulation of E-cadherin, represents one of the possible mechanisms of EMT involvement in cancer development.

Activation of EMT by different stimuli or stressors can generate cells of intermediate or hybrid states of the broad spectrum between epithelial and mesenchymal (E/M), which are interchangeable [54]. In fact, a recent study by Kröger [64] and colleagues demonstrated that tumorigenicity of selected highly tumorigenic breast cancer cells depended on cells lying on an intermediate E/M state, rather than at any extreme of the spectrum. This may explain the reason why hybrid phenotypes are associated with poorer patient prognosis [7]. In addition, activation of EMT induces stemness in non-SCC. Induction of EMT in immortalized human epithelial cells, either by ectopic expression of Twist or Snail or by exposure to TGF-β, resulted in acquisition of a CSC, CD44^high^/CD24^low^, phenotype and increased mammosphere formation ability [65]. Similarly, in prostate cancer, the EMT phenotype was linked with increased stemness as demonstrated by upregulation of pluripotency markers Sox2, Nanog, Oct4, Lin28B and Notch1, increased sphere formation ability and enhanced tumorigenicity in mice [66]. Overexpression of Snail in colorectal cancer promoted a CSC-like phenotype, with improved migration, invasion and metastasis formation [67]. Moreover, expression of Zeb1 in breast non-SCC increased upon microenvironment signaling, subsequently leading to a CSC state [8]. As mentioned above, the stimulus to undergo EMT may arise from cells of the microenvironment such as CAF or TAM but also from CSC in a positive-feedback fashion, increasing their density in the tumor. Accordingly, CD133^+^ ovary CSC could induce EMT in CD133^-^ non-SCC and increase the metastatic capacity of those cells in vivo and in vitro [68].

EMT and stemness seem important for tumor invasion and metastization. Upon EMT activation, tissue evasion and intravasation are promoted [69], as cells positive for EMT markers localize preferentially closer to blood vessels [70], similarly to putative CSC, facilitating the access to the blood stream. After leaving the blood stream to the secondary organ, the mesenchymal-like state is reversed and the metastasis is established [69].

Altogether, from the data collected so far, CSC and the tumor microenvironment suffer mutual influences. Stemness seems more like a transient dynamic property of cancer cells, modulated during tumor development, other than a stationary characteristic of a specific subpopulation of cells. Notwithstanding, the recognition of the importance of EMT on CSC and non-SCC duality has a relevant impact on the understanding of tumor biology, and on therapeutic approaches and potential drug resistance.

### 3.3. From Resistance to Standard Therapy to Stemness-Based Therapeutic Intervention

According to current concepts, CSC are involved in drug resistance and subsequent tumor relapse. CSCs are pointed as naturally resistant to chemo- and radiotherapies due to several intrinsic mechanisms. Those include increased DNA repair capacity after exposure to radiation [71]; high expression of efflux bombs like the ATP-binding cassette transporters [72]; high expression of antiapoptotic/survival pathways [73] and ALDH [41]; high levels of free-radical scavengers, which reduce intracellular reactive oxygen species (ROS) [74], and high self-renewal capacity and quiescence [41].

Exposure of non-SCC to chemotherapy agents seems to promote adaptive therapy resistance upon gaining of stemness. In 2010, Sharma and colleagues [75] identified a subpopulation of cancer cells with a dynamically acquired transient drug-resistance phenotype established by IGF-1R signaling and a specific metastable chromatin state, independent of drug efflux. Later, it was demonstrated that exposure to carboplatin could induce self-renewal and pluripotency of hepatocellular non-SCC, suggesting these cells as a source of CSC, representing a relevant mechanism for therapy resistance [76]. Similarly, in triple negative breast cancer, paclitaxel and carboplatin promoted a CSC enrichment mediated by HIF-1 [77]. Furthermore, after exposing ovarian cancer cell lines to cisplatin, doxorubicin or paclitaxel, a reversible increase in CXCR4^high^/CD24^low^ CSC population was reported [78]. Moreover, in ovarian cancer, treatment with cisplatin or carboplatin increased IL-6 secretion by cancer-associated fibroblasts, which promoted enrichment of ADLH^+^ CSC [79], consistent with IL-6 mediated EMT activation in gastric cancer [80] and a tight correlation between CSC plasticity and EMT. Resistance to afatinib, an effective EGFR-tyrosine kinase inhibitor, was correlated with EMT features and putative stemness in several cell lines [81] and further confirmed in a patient case-report [82]. Dysregulation in CSC metabolism is another valid mechanism for acquired stemness and therapy resistance [83].

Considering the growing evidence defining stemness in cancer cells as an important and adaptable source for therapy resistance, targeting CSC is a logic and important strategy to overcome the problem and prevent relapse (Figure 3). Therapies directed against CSC may encompass targeting stemness and EMT pathways (reviewed elsewhere [9,84,85]) and/or specific cell surface proteins/receptors [14,86,87], which identification remains as a crucial challenge.

Several drugs targeting CSC/EMT pathways, developmental pathways and surface receptors are currently undergoing clinical trials (at different stages) [88]. Those include drugs targeting Wnt pathways such as porcupine (PORC) inhibitors (e.g., WNT-974, ETC-159) and β-catenin inhibitors (e.g., BC-2059), NOTCH pathway, such as AL-101 γ-secretase inhibitor or Hedgehog pathway, such as Patidegib or Taladegib. Drugs targeting either surface markers, such as CD44, or EMT pathways [85] are also undergoing clinical trials (extensively reviewed in Yang et al. [89]). Other trials, including bevacizumab against ovarian cancer or dasatinib against prostate cancer include EMT evaluation as an endpoint [85]. Actually, Vismodegib (p-glycoprotein inhibitor) and Sonidegib phosphate (Smo receptor antagonist) were launched in 2012 and in 2015, respectively, for the treatment of basal cell carcinoma [88]. Furthermore, several Chimeric Antigen Receptor T (CAR-T) cells are under development against putative CSC markers, including EpCAM [89].

Ongoing research aims at identifying new possible targets. Anti-CDH11 antibodies are a promising therapeutic strategy for the treatment of metastatic breast cancer as it reduces EMT and CSC-like features [90]. In non-small cell lung cancer cell lines, targeting CD133^+^ CSC with TNF-related apoptosis inducing ligand (TRAIL) seems to induce CSC apoptosis [91]. Compounds that revert EMT are showing themselves effective in overcoming chemotherapy resistance in in vitro and in vivo models [92,93].

It is apparent that current development of therapeutic strategies towards CSC has been focused on targeting specific stemness processes and signaling pathways, like EMT, or putative surface markers. Nonetheless, the plastic nature of CSC/non-SCC constitutes a challenge to such strategies as cells may shift their dependence on those processes, thus evading treatment. Accordingly, one should envision a broader strategic intervention when considering tackling CSC, potentially exploring a common marker expressed in different compartments of the tumor microenvironment.

In this respect, nucleolin, overexpressed in the tumor vasculature [11,94] and cells of certain tumors (e.g., breast), has been shown to be present in both non-SCC and CSC, thus representing a wider therapeutic marker for targeted therapeutic intervention at the tumor microenvironment, including the CSC niche [14,15,84,95,96].

## 4. Multifunctional Protein Nucleolin—A Possible Driver of the Cancer Hallmarks?

### 4.1. Structure and Localization

In 1973, Orrick and colleagues [97] separated nucleolar proteins from normal rat liver cells, and Novikoff hepatoma ascites in a two-dimensional polyacrylamide gel electrophoresis. This work enabled the identification of nucleolin, for the first time, as C23 protein, one of the most abundant nucleolar proteins. Human nucleolin is a protein encoded by the NCL gene localized on the short arm of chromosome 2 [98], and consists of 717 amino acids [99] of nearly 77 kDa of molecular weight. However, due to post-translational modifications the measured molecular weight is, approximately, 100–110 kDa [100].

Nucleolin is a highly conserved phosphoprotein, present in several species, from yeast and plants to animals. Seventeen conserved dibasic cleavage sites have been identified on its sequence, possibly to generate new nucleolar proteins [101]. The structure of nucleolin consists of three different functional domains (Figure 4): (1) a N-terminal with acidic stretches intercalated with basic regions and a nuclear localization signal (NLS), (2) a central domain with four RNA-binding domains (RBD) or RNA recognition motifs (RRMs) and (3) a C-terminal with glycine and arginine rich (GAR) domains, also enriched in phenylalanine [10].

The N-terminal of nucleolin varies in length among species. The acidic stretches that characterize this region have been proposed to bind histone H1 leading to chromatin decondensation [102], which can possibly influence ribosomal DNA (rDNA) transcription [103]. Several phosphorylation sites have been identified within this particular domain [102], also associated with protein–protein interactions [10]. This is highly suggestive of the existence of multiple protein partners for nucleolin. The central domain possesses five potential N-glycosylation sites [99]. The RBDs vary in number depending on its origin, being animal nucleolin (including human) characterized by four RBDs [10]. These domains allow specific interactions with nucleic acids such as mRNA and rRNA [104], participating in pre-rRNA processing [103]. Like the N-terminal, the GAR domain of the C-terminal varies in length among species and is, as well, a protein–protein and RNA interaction domain. It interacts mainly with ribosomal proteins and possibly facilitating nucleolin contact with large and complex RNAs such as rRNA [10].

The largest cellular pool of nucleolin is in the nucleolus. In mammalian cells, nucleolus is characterized by three main components: the dense fibrillar component, a major reservoir of nucleolin, surrounded by the fibrillar centers and both embedded in the granular component [105]. Nucleolin has also been identified in some extension in the granular component, but rarely in the fibrillar centers [10]. Furthermore, nucleolin has been found to constantly shuttle back and forward between the nucleus and the cytoplasm [106] especially due to the presence of a bipartite nuclear localization signal (NLS). In addition to the NLS, the RBD and GAR domains are required for the accumulation of nucleolin in the nucleolus [107]. Phosphorylation is necessary for nucleolin nuclear-to-cytoplasm shuttling, as cytoplasmic nucleolin overlapped with extensive phosphorylation, while dephosphorylation overlapped with nuclear translocation [108]. This shuttling is argued to be relevant for the transportation of molecules from one cell compartment to the other, especially of core ribosomal proteins, which interact with the nucleolin GAR domain [109].

Although nucleolin has no identifiable transmembrane domains, it has been detected on the cell surface interacting with other molecules [10]. N-glycosylation seems to be necessary for the translocation of the protein to the membrane [110,111], which is relevant from a therapeutic point of view.

### 4.2. Role of Nucleolin on Ribosomes Biogenesis, Gene Transcription and Translation

The preferential accumulation of nucleolin in the transcriptionally active nucleolus and its association with rRNA and other nucleic acids, underlies the relevance of this protein for ribosome biogenesis and gene expression. Actually, nucleolin appears to modulate chromatin condensation during rDNA transcription initiation through interaction with histone H1 [10]. Furthermore, the maturation of a 100 kDa protein physiochemically equivalent to nucleolin was associated with rDNA transcription [112] and with exponentially proliferating cells [113], concordant with the role of nucleolin on the regulation of these processes. Both studies pointed nucleolin as a crucial protein for ribosome maturation [112,113]. Moreover, it has been controversially implicated in the regulation of RNA polymerase I (RNAPI). For instance, microinjection of antinucleolin antibodies [114] or of 2-4-fold excess of nucleoli [115] were associated with repression of RNAPI activity. On the other hand, nucleolin knock-down or RNA interference models pointed for decreased RNAPI activity, suggesting a role of nucleolin on the activation of the enzyme [116].

Through its RBDs, nucleolin interacts at a very early stage with specific structures of nascent rRNAs [117], named nucleolin recognition elements and evolutionary conserved motifs [118]. These interactions establish a ribonucleoprotein complex required for the first step of pre-rRNA processing and cleavage [119,120] and for the correct folding of rRNA and ribosomes assembly [118], suggesting nucleolin as an RNA chaperone. Nucleolin has also been found to interact with several mRNAs (see Table 2), influencing their translation and protein levels in both normal physiology and disease.

Transcriptome-wide studies revealed that nucleolin binds specific G-rich sequences in the coding region and untranslated regions (UTRs) of target mRNAs, enhancing or repressing (e.g., p53) their translation. Most of the identified mRNA targets of nucleolin encode proteins associated with cellular growth and proliferation, many of which are related to cancer development [130].

### 4.3. Nucleolin as a Regulatory Protein of Proliferation, Cell Cycle and Cell Survival

The levels of nucleolin expression are directly associated with the rate of cell proliferation and subsequently cell cycle rates [131]. Expression of nucleolin increases by 3.5-fold as the cell cycle is induced [132], suggesting a role for nucleolin in the transition of growing cells through the cell cycle phases. In fact, if nucleolin is absent, cells will experience a mitotic delay due to the activation of the spindle-checkpoint owing to an improper kinetochore-microtubule attachment and consequent misalignment or non-aligned chromosomes [133], especially since nucleolin is a core component of the centrosome. It is implicated in the maturation of the centriole and activation of microtubule nucleation, presumably after binding to centrosomal γ-tubulin ring complexes (γTuRC). Therefore, nucleolin is crucial for the organization of the microtubules network and their attachment to centrosomes [134], and for maintaining the integrity of the nucleolus [133]. This was reinforced by Ugrinova and colleagues [135], who demonstrated cell growth arrest of HeLa cells, leading to an accumulation in the G2 phase and defects in centrosome duplication. This resulted in multipolar spindle formation upon siRNA-mediated downregulation of nucleolin [135]. Wang et al. (2014) [136] demonstrated inhibition of proliferation of human umbilical vein endothelial cells (HUVEC) through cell cycle arrest in the S phase, upon nucleolin downregulation mediated by ADP treatment. In addition, nucleolin was implicated in the regulation of vascular smooth muscle cells proliferation in atherosclerosis via protein–protein interaction with Aurora B [137].

Post-translational modifications of nucleolin may significantly influence its regulatory functions in cell cycle. For instance, nucleolin phosphorylation of the N-terminal extensively occurs during interphase by CK2, and during mitosis by cdc2 [138].

Nucleolin may also bias cell proliferation through binding and stabilization of anti-apoptotic BCL-2 and BCL-XL mRNAs, preventing their degradation and enhancing their translation, which might reduce apoptosis [123,124]. Accordingly, the downregulation of nucleolin promoted increased apoptosis of HeLa cells [135] and embryonic stem cells (ESC) [139]. Nonetheless, the role of nucleolin in stemness, pluripotency and cell differentiation remains underexplored.

### 4.4. Nucleolin in Tumor Initiation and Progression

Nucleolin was found to be overexpressed in several types of tumors such as melanoma [140], leukemia [141], gastric cancer [142], glioma [143], colorectal cancer [144], hepatocellular carcinoma [12], ependymoma [145], lung cancer [13] and breast cancer [146]. Some of these studies positively associated higher levels of nucleolin with worse prognosis in leukemia [141], glioma [143], hepatocellular carcinoma [12] and breast tumors [147]. In gastric cancer [142] and non-small cell lung cancer [13], nucleolar nucleolin seems to be a marker of better overall survival while its accumulation in the membrane or in the cytoplasm is associated with increased malignancy grade and poorer outcome.

Nucleolin is implicated in diverse cellular processes in physiological conditions, which are typically dysregulated in tumors, such as proliferation and cell cycle, apoptosis evasion and angiogenesis. For instance, nucleolin is highly expressed in actively dividing cancer cells [131], promotes the stabilization of antiapoptotic mRNAs [123,124] and blocks the proapoptotic Fas receptor [148]. If located in the membrane, it is involved in tumor angiogenesis [149], binding of ligands involved in tumorigenesis [150] and stemness maintenance of breast cancer cells [14].

As summarized in Figure 5, nucleolin was found to interact with the cytoplasmatic domain of ErbB family of tyrosine kinase receptors (such as EGFR), inducing the phosphorylation, dimerization and activation of these receptors [151], presumably through Ras [152]. EGFR activation by nucleolin triggers intracellular MAPK signaling cascade, thus promoting ligand-independent cell growth [146]. Moreover, nucleolin has been implicated in the mediation of the carcinogenic effects of certain factors through the activation of Erk and PI3K-Akt pathways [153,154]. Upon Akt activation through EGFR stimulation, nucleolin is phosphorylated and increases the levels of the transcription factor Sp1 involved in cell cycle, apoptosis, differentiation and tumorigenesis, by both facilitating translation and protein stabilization [155]. Additionally, p85α mediates nucleolin transcription and expression, subsequently stabilizing EGFR mRNA, increasing its protein levels and promoting cell malignization [122].

Nucleolin also regulates the expression of microRNAs (miRNAs) namely miR-21, miR-221, miR-222 and miR-103, whose overexpression is associated with breast cancer initiation, metastases and therapy resistance [156]. Similarly, in prostate cancer cell lines, targeting and inhibition of nucleolin promoted a decrease in the level of those same oncogenic microRNAs and impaired cell proliferation and migration [157].

In head and neck squamous cell carcinoma, nucleolin was suggested as a mechanism of adhesion to lymphocytes via L-selectin in a shear-dependent manner. More specifically, nucleolin expressed on the surface of cancer cells actively bind to L-lectin [158].

### 4.5. Cell Surface Nucleolin and Interaction with External Ligands

Although nucleolin is typically a nucleolar protein, it has been also detected on the cell surface as a part of the shuttling mechanism [10]. Cell surface nucleolin has been identified in cancer cells and angiogenic blood vessels [11,159], and implicated in angiogenesis and apoptosis [149,160,161]. The mechanisms through which nucleolin is translocated to the membrane are not fully clarified. Nonetheless, a N-glycosylation seems necessary for this translocation to occur [111]. Moreover, under vascular endothelial growth factor (VEGF) stimulation, myosin heavy chain 9 (MyH9) serves as a linker between nucleolin and cytoplasmic actin filaments providing an anchorage to the cell surface [160]. Heat shock cognate 70 (Hsc70) has been shown to interact with nucleolin and mediate its relocation to the surface through enhanced stability and interaction with MyH9. Phosphorylation of nucleolin by protein kinase C ξ (PKC-ξ) and CK2 highly influence its binding to Hsc70 [161].

Nucleolin at the cell membrane works as a functional cell surface protein for endostatin-mediated antiangiogenic activity [162], and also as a ligand of the proapoptotic Fas receptor preventing apoptosis [148]. Blocking cell surface nucleolin with specific antibodies increases apoptosis and decreases migration of endothelial cells and prevents angiogenesis [149,160]. Being present at the cell surface, nucleolin interacts with numerous ligands and even mediate the internalization of nanoparticles and viruses [163] (Table 3).

Overall, nucleolin has been identified on the surface of cancer and tumor endothelial cells and has been implicated in diverse cellular processes, making it an appealing target for anticancer therapies.

## 5. Role of Nucleolin on Stemness, Pluripotency and Differentiation: A Potential Target for Broad Anticancer Therapy

The multiple functions of nucleolin explored above dictate critical relevance for cell biology, either physiologically or pathologically. However, its relevance for stemness (i.e., embryonal development or CSC) has remained rather illusive. Nevertheless, clues from different studies are starting to build our understanding of nucleolin in stem-like states. An important aspect stressing nucleolin’s critical physiological relevance is the absence of any described viable knockout mouse model, which points it as of utmost importance on embryonal development. Indeed, it has been shown that phosphorylated nucleolin interacts with Tpt1 (translationally controlled tumor protein) in murine ESC, a complex that, while increasing during mitosis, is reduced upon cell differentiation. Furthermore, it also interacts with the transcription factor Oct4, at interphase, in both human and murine ESC [172]. In fact, the work of Yang and colleagues [139] indicates that nucleolin interferes with the regulation of the ESC self-renewal ability as it is highly expressed in these cells and its downregulation induces cell differentiation, in a p53-dependent manner. This has been previously suggested by Takagi et al. (2005) [121], who have shown that nucleolin negatively regulated p53 translation. Moreover, nucleolin, together with LINE1 retrotransposon and Kap1, was shown to repress Dux, the master activator of a transcriptional program specific to the 2-cell embryo, enabling 2-cell embryo exit towards the embryonic stem cell state, while promoting ES self-renewal [173]. Further evidence supporting stemness functions is related to nucleolin’s importance in the activation of CD133, a marker of hematopoietic stem/progenitor cell (HSPC), enabling the increase in colony-forming units and promoting the long-term maintenance of hematopoietic progenitor cells [174], upon activating Wnt/GSK3β signaling [175].

Altogether, these studies demonstrate that nucleolin is a pivotal regulator of stemness programs and an essential contributor to embryonal development. Accordingly, if one accounts that cellular reprograming is an event that may occur in many cancers (for instance, through EMT), which constitutes an opportunity for gain of functions, the overexpression of nucleolin could essentially function as one of the drivers of stem-like features of CSCs. Some studies have already started to unveil the importance of nucleolin in CSC biology, although not in the same extent in ES and HSPC.

Indeed, in a colon cancer cell line, Caco2/TC7, laminin-1 (Ln-1)-induced differentiation, displaced nucleolin from the nucleolus to the cell membrane. In this case, nucleolin shRNA-mediated downregulation mimicked the differentiation process [176]. Furthermore, a positive association between pluripotency markers and nucleolin mRNA levels was found in TNBC cells and in ESC. Upregulation of Nanog and Oct4 in sorted ALDH^high^/CD44^high^ TNBC stem cells was accompanied by an upregulation of nucleolin. Additionally, ESC cultured in conditions enabling pluripotency, displayed similar upregulation of Nanog and Oct4 accompanied by nucleolin upregulation [14]. Importantly, in limiting the dilution functional assay it was shown that sorted nucleolin^+^ triple negative breast cells were more tumorigenic in NOD scid gamma mice than nucleolin^low/-^ cells, evidencing that cell surface nucleolin *per se* enables the selection of efficient tumorigenic cells [14].

Interestingly, treatment of neuroblastoma cells with Roniciclib, which induced cell differentiation and impaired neurospheres formation, also inhibited the expression of nucleolar nucleolin and of CSC markers such as CD44v6 and CD114 [177].

Aggregating all the described features and implications for stemness and carcinogenesis, one may envision nucleolin as a relevant marker for targeted anticancer therapies due to its presence at the cell membrane of multiple cellular subpopulations of the tumor microenvironment, including CSC [178]. In this respect, several strategies exploring cell surface nucleolin are under development.

For example, the AS1411 aptamer is an antiproliferative G-rich phosphodiester oligonucleotide, which binds specifically to cell surface nucleolin and further internalizes [179]. This aptamer successfully produces antitumorigenic effects by decreasing the levels of nucleolin-related miRNAs [156], disrupting the antiapoptotic pathway NK-kB [180] and impairing BCL-2 mRNA stabilization [181]. Similarly, the pseudopeptide HB19, which especially binds the GAR domain of cell surface nucleolin, produces an antagonist effect as it promotes antitumorigenic effects like arresting of tumor growth and angiogenesis [182]. For this reason, nucleolin has been considered for nanotechnology-based targeted delivery of chemotherapy agents. For instance, the F3 peptide, which has binding specificity for nucleolin [11], has been used to functionalize liposomal formulations for delivery of encapsulated single [15], nucleic acids [183,184] or combination of anticancer drugs [14,185] to non-SCC and CSC, associated with marked antitumor effects [15]. Furthermore, the F3 peptide was engineered as part of a modular construct, F3-RK-PE24-H6 (containing the cationic peptide (RK)n, and the toxin domain PE24 of *Pseudomonas aeruginosa*), shown to assemble as discrete nanoparticles toxic to its target cells, triple negative breast CSC [186].

More recently, AGM-330 peptide was demonstrated to specifically bind nucleolin on the surface of cancer cells in vivo and in vitro. The conjugation of paclitaxel with AGN-330 improved cancer cell growth compared to treatment with paclitaxel alone [187].

Altogether, taking the fact that nucleolin is both expressed in CSC and non-SCC, and in other cell compartments as the tumor vasculature, one might envision nucleolin as a pan-target, that when explored to promote the delivery of the adequate drugs, may enable the debulk of tumors while simultaneously providing a mean to precisely tackle CSC, the embodiment of stemness in cancer.

## 6. Conclusions

Cells with stem-like phenotypes have already been identified in a huge variety of cancers. These cells are very often called the CSC. However, they are not a well-defined and static population of tumor cells. Stemness is rather a transient adaptive property of cancer, which may be activated, for instance, to initiate metastasis, or lost by differentiation to establish a heterogeneous tumor. One should then consider the inclusion of stem-like phenotypes as an emerging hallmark of cancer, which can be linked with the already established hallmarks.

Nucleolin dysregulation is clearly driving cancer cells into aberrant states related to the hallmarks, including sustained proliferation, promotion of angiogenesis, escaping of apoptosis and even tissue evasion. A role in the maintenance of stemness has also been suggested but not completely understood, which could be perused in future research, namely in the development of stemness-targeted therapies.

## Figures and Tables

**Figure 1 pharmaceuticals-14-00060-f001:**
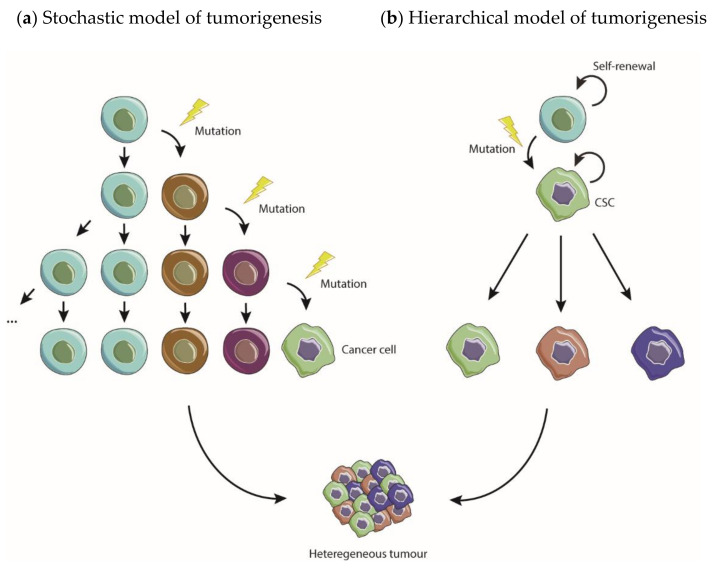
Stochastic and hierarchical models of carcinogenesis. (**a**) According to the stochastic model, all cancer cells have equal ability to initiate tumors and their evolution depends exclusively on stochastic influences. (**b**) The hierarchical model postulates that cancer stem cells (CSCs) are the only ones able to initiate, maintain and seed new tumors.

**Figure 2 pharmaceuticals-14-00060-f002:**
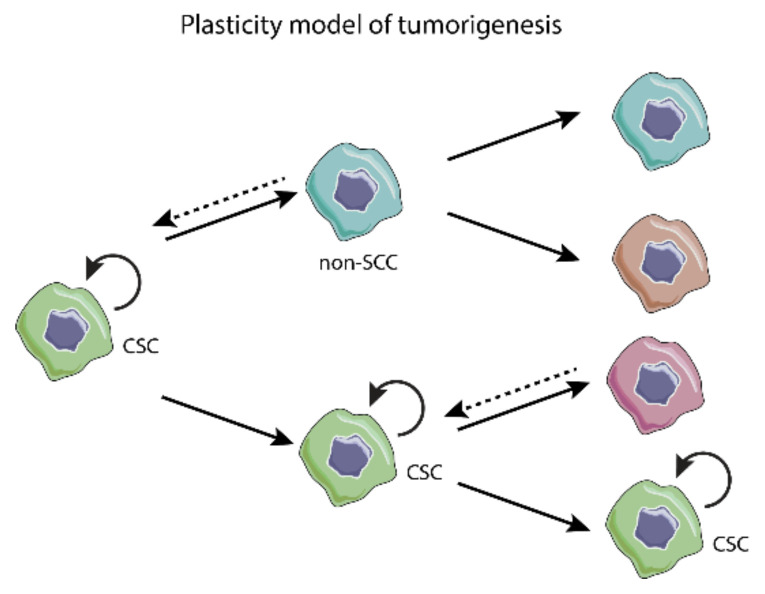
Plasticity model of tumorigenesis. Cancer stem cells (CSCs) can differentiate and originate any cell type of the tumor. In turn, non-stem cancer cells (non-SCCs) are able to alter their state toward a stem-like phenotype in response to physiological stimuli.

**Figure 3 pharmaceuticals-14-00060-f003:**
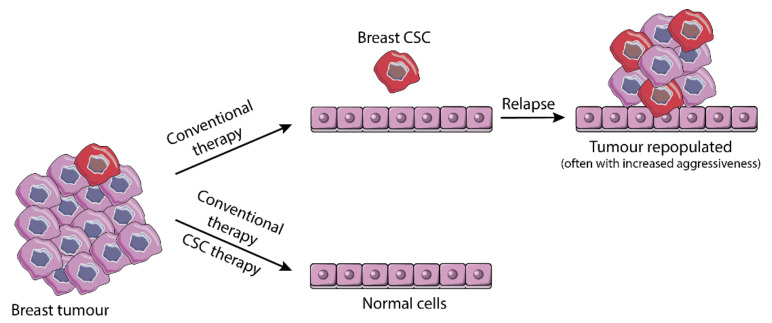
Schematic representation of cancer stem cell (CSC)-mediated tumor relapse. CSCs are resistant to conventional chemo- and radiotherapy. The remaining quiescent population of CSC can regain activity and initiate a new tumor, leading to relapse. Therapies targeted to CSC aim to eliminate this subpopulation and consequently reduce the risk of relapse.

**Figure 4 pharmaceuticals-14-00060-f004:**
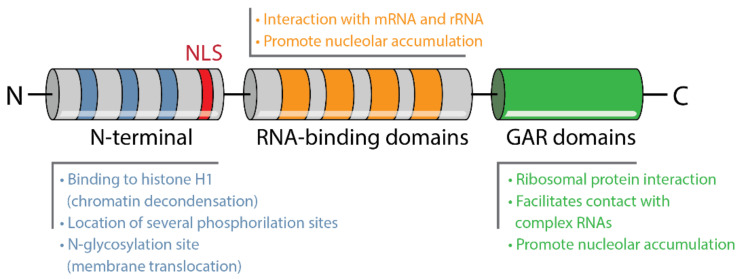
Schematic structure of nucleolin with acidic stretches on N-terminal, nuclear localization signal (NLS), RNA-binding domains in the central region and glycine and arginine rich (GAR) domains in the C-terminus, and their related major functions or properties.

**Figure 5 pharmaceuticals-14-00060-f005:**
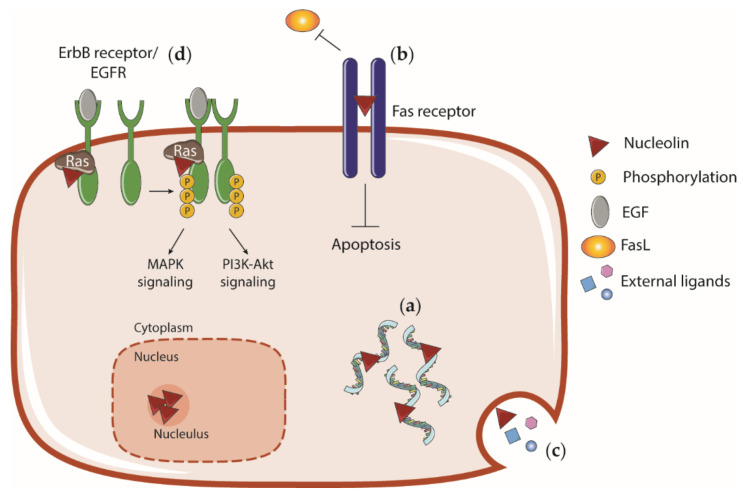
Role of nucleolin in tumorigenesis. (**a**) Nucleolin is overexpressed in a variety of tumors and stabilizes tumorigenic mRNAs that codify, for instance, antiapoptotic proteins or EGFR. It is also able to induce the translation and stabilization of tumorigenic proteins. (**b**) At the cell surface, nucleolin blocks the proapoptotic receptor Fas inhibiting its interaction with FasL and (**c**) binds external ligands involved in tumorigenesis and angiogenesis mediating their internalization. (**d**) Nucleolin binds to EGFR (through Ras) and promotes phosphorylation, dimerization and activation of the receptor leading to Erk/MAPK or PI3K-Akt pathways signaling, both independently or after EGF stimulation.

**Table 1 pharmaceuticals-14-00060-t001:** Putative cancer stem cell markers for tumors of diverse histological origin.

Tumor	Markers	References
Acute Myeloid Leukemia	CD34^+^/CD38^−^	[6,21]
Breast	CD44^high^/CD24^low^	[20]
Ovary	CD24^+^, CD133^+^, CXCR4^+^	[25,42]
Ewing’s Sarcoma	CD133^+^	[29]
Lung	CD133^+^, CD90^+^	[43]
Prostate	CD44^+^/α2β1integrin^high^/CD133^+^	[44]
Colorectal	CD44^+^, CD133^+^, CD166^+^, Lgr5^+^, EpCAM^+^	[45]
Pancreas	CD44^+^, CD24^+^, EpCAM^+^	[46]
Brain	CD90^+^, CD133^+^	[47,48]
Melanoma	CD271^+^	[49]

Putative cancer stem cells may be identified by surface markers dependent on the histological origin of the tumor. CD: Complex of Differentiation; CXCR4: C-X-C chemokine receptor type 4; EpCam: Epithelial Cell Adhesion Molecule; Lgr5: Leucine-rich repeat-containing G-protein coupled receptor 5.

**Table 2 pharmaceuticals-14-00060-t002:** Interaction of nucleolin with mRNAs.

Interaction with	Attributed Functions/Impact	References
5′UTR of *p53* mRNA	Suppression of p53 translation and induction after DNA damage	[121]
*EGFR* mRNA	Stabilization of EGFR mRNA and increased expression of the receptor involved in cell malignization	[122]
AU-rich element of *BCL-2* mRNA	Stabilization of *BCL-2* mRNA and decreased apoptosis	[123]
*BCL-X_L_* mRNA (when phosphorylated)	Stabilization of *BCL-X_L_* mRNA and decreased apoptosis	[124]
3′UTR *APP* mRNA	Stabilization of *APP* mRNA and consequent accumulation of *APP* protein in Alzheimer’s disease	[125]
Kinesins and importin β1 mRNA	Transportation of importin β1 mRNA to specific sites in cells to control cell growth	[126]
Selenoproteins mRNA	Selective enhancing of a subset of selonoproteins at the level of translation	[127]
*IL-2* mRNA	Stabilization of IL-2 mRNA during T-cells activation	[128]
*COX-2* mRNA	Stabilization of *COX-2* mRNA leading to COX-2 upregulation and consequent malignant transformation	[129]

Nucleolin interacts with a variety of mRNAs with different results. APP: Amyloid Precursor Protein; COX-2: Cyclooxygenase 2; Epidermal growth factor receptor; EGFR: IL-2: Interleukin-2; UTR: Untranslated Region.

**Table 3 pharmaceuticals-14-00060-t003:** External ligands of nucleolin.

Ligands	Attributed Functions/Impact	References
F3 peptide (synthetically derived from HMGN2)	Targeting of tumor endothelial cells and tumor cells; possible deliverer of therapeutic molecules.	[11,14,15,94]
Urokinase	Formation of a complex that includes nucleolin, urokinase receptor and CK2 that mediates the mitogenic activity of urokinase.	[164]
Lactoferrin	Internalization of lactoferrin and induction of recycling/degradation pathway or nucleolus translocation.	[165]
P-selectin	Interaction with P-selectin on the cell surface of human colon carcinoma cells and formation of a signaling complex that includes phosphorylated surface nucleolin, PI3K and p38 MAPK. This complex regulates cell adhesion and spreading which are implicated in carcinogenesis.	[150]
LPS	Internalization of LPS on activated alveolar macrophages and consequent mediation of the inflammatory response to bacterial infection.	[166]
Apoptotic cells	Interaction of macrophage surface nucleolin with apoptotic cells signalized to phagocytosis.	[167]
Influenza A viruses	Internalization of several subtypes of influenza A viruses thus mediating infection.	[168]
Tipα	Internalization, on gastric cancer cells, of Tipα (carcinogenic factor of *Helicobacter pylori*).	[169]
Enterovirus 71	Mediation of enterovirus 71 cell infection.	[170]
Respiratory syncytial virus	Interaction with respiratory syncytial virus at the apical membrane and mediation of infection.	[171]

Nucleolin interacts with a variety of external ligands with different results. CK2: Casein Kinase 2; LPS: Lipopolysaccharide; MAPK: MAP Kinase.

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
