# Peer review of "Cancer Stem Cells and Nucleolin as Drivers of Carcinogenesis"

_pharmaceuticals, 2021, doi:10.3390/ph14010060_

Round 1

Reviewer 1 Report

The authors have written a comprehensive review on cancer stem cells and use of nucleolin as a targeted anti cancer therapy. There is a lot of information in the review and it is well organized and presented. I have few suggestions:

  1. The introduction needs to include more details on cancer stem cells and a rationale for writing a review on them. Also, authors can include other proteins like nucleolin that are successful as anti cancer therapeutics either due to their location on cell membrane or due to their anti tumor function. They can refer readers to other reviews and then explain what to expect in this review. 
  2. In the review, the authors should include some sentences about their inference on various studies. It is not sufficient to just list various studies and findings. It is also important to consolidate and reflect on studies that have been published. 
  3. In figure 4, authors can also show various functions of nucleolin domains in the cartoon.
  4. The authors can discuss more the role of immune system in cancer stem cell evasion.
  5. Also, it will be useful to talk about other drugs that target CSC and are either in late stage trials or in use to treat cancer. 

Reviewer 2 Report

Figure 3: I think the authors should reconsider what they understand by secondary tumors

https://www.cancer.gov/publications/dictionaries/cancer-terms/def/secondary-tumor

secondary tumor: a term used to describe cancer that has spread (metastasized) from the place where it first started to another part of the body. Secondary tumors are the same type of cancer as the original (primary) cancer. For example, cancer cells may spread from the breast (primary cancer) to form new tumors in the lung (secondary tumor). The cancer cells in the lung are just like the ones in the breast. Also called secondary cancer.

Line 411: two signaling cascades should be cited: PI3k-Akt and ERK/MAPK. The sentence, as written by the authors, seems to indicate that there are three pathways.

Why nucleolin and CSC have been talked about in this review? Why not only nucleolin? It would have been more focused. Now, it is like joining two different concepts together.

The authors speak about heterogeneity of tumors, based on the models of cancer development. But they do not speak about clonality. It would be interesting that they cite different articles explaining that tumors are monoclonal although heterogeneous (cellular and molecularly speaking).

When CSC are described, ABC transporters are mentioned, but not described, and they are the cause of chemoresistance. More description on ABC transporters is needed, together with their different classes (MRP, MDR, ABCG2…). Also the concept of cells of the side population (SP) should be discussed (it is related to the proved existence of ABC transporters), and also related to the concept of CSC, although we cannot say that cells of the SP are equal to CSC. A discussion on those points would be interesting.

The pathways of CSC, the genetics of them, are not described: sonic hedgehog, wnt, etc. It is important to understand the molecular genetics of CSC and why we should point to treat CSC targeting those pathways.

Line 101: the title says: 3.1. Models of tumorigenesis – a new paradigm driven by CS. Maybe there is a mistake on CS (some letter omitted).

Fig 2: non-SSC. It is maybe a mistake.

EMT is described, but the authors should also include the reversal (MET) in their discussion.

Point 4.7 is rather shortly discussed, given its importance. Trying to increase this part of the review would be better.

In general, the review is written in a way that it is a bit hard to read, as many citations are incorporated just to say a short phrase of each of them. A certain lack of personal view is seen in the way the review is written. The authors should increase their personal when writing about the topic; they should not only incorporate lots of citations. If they follow this recommendation, the review will gain much and will be easier to read.

Reviewer 3 Report

This review covers cancer stem cell markers and concepts (2,3), general introduction of nucleolin(4.1-4.3, 4.5), and finally nucleolin’s role of cancer cell stemness(4.4  shortest). Each section is informative in its shape. The title, however, highlights nucleolin in the light of cancer cell stemness, which only the section 4.4 is relevant.

I would suggest the authors to expand section 4.4 and introductory and/or transitional section would be necessary to lead the readers from cancer stem cells to nucleolin.

If the author have any real image (IHC or IF) of translocation of nucleolin to the membrane, the readers would enjoy more.

Round 2

Reviewer 1 Report

The authors have addressed my concerns and I support publication at this time. 

Reviewer 2 Report

All changes are appropriate and of benefit for the manuscript.